# SARS-CoV-2 VOC type and biological sex affect molnupiravir efficacy in severe COVID-19 dwarf hamster model

Carolin M. Lieber[1,5], Robert M. Cox [1,5], Julien Sourimant [1], Josef D. Wolf [1], Kate Juergens[2], Quynh Phung[2], Manohar T. Saindane[3], Meghan K. Smith [3], Zachary M. Sticher [3], Alexander A. Kalykhalov[3], Michael G. Natchus[3], George R. Painter[3], Kaori Sakamoto[4], Alexander L. Greninger[2] & Richard K. Plemper [1] ✉

SARS-CoV-2 variants of concern (VOC) have triggered infection waves. Oral antivirals such as molnupiravir promise to improve disease management, but efficacy against VOC delta was questioned and potency against omicron is unknown. This study evaluates molnupiravir against VOC in human airway epithelium organoids, ferrets, and a lethal Roborovski dwarf hamster model of severe COVID-19-like lung injury. VOC were equally inhibited by molnupiravir in cells and organoids. Treatment reduced shedding in ferrets and prevented transmission. Pathogenicity in dwarf hamsters was VOC-dependent and highest for delta, gamma, and omicron. All molnupiravir-treated dwarf hamsters survived, showing reduction in lung virus load from one (delta) to four (gamma) orders of magnitude. Treatment effect size varied in individual dwarf hamsters infected with omicron and was significant in males, but not females. The dwarf hamster model recapitulates mixed efficacy of molnupiravir in human trials and alerts that benefit must be reassessed in vivo as VOC evolve.

By May 2022, SARS-CoV-2 has resulted in over 522 million cases and >6.2 million deaths worldwide[1]. Vaccines are widely available[2,3], but recuring global infection waves have been fueled by limited longevity of vaccine-induced immunity, the hesitancy of population subgroups to vaccinate[4,5], and increasingly contagious and/or vaccine-insensitive variants of concern (VOC) alpha (B.1.1.7 lineage), beta (B.1.351 lineage), gamma (P.1 lineage), delta (B.1.617.2 lineage), and omicron (B1.1.529 lineage)[6,7]. VOC delta was the prevalent circulating variant during Summer and Fall 2021 due to replication to high titers, prolonged shedding from infected individuals, and propensity to induce breakthrough infections in vaccinees[8–10]. Since its first appearance in November 2021, VOC omicron has rapidly replaced delta as the dominant circulating strain in most geographical regions[11], propelled

by sharply reduced sensitivity to neutralizing antibodies directed against earlier lineages and greatly increased infectivity[12]. Although clinical signs associated with VOC omicron are typically milder than those of its predecessors, record-high daily infection rates have driven high absolute hospitalization numbers, creating an urgent need for therapeutics to improve disease management.

Molnupiravir was the first orally available SARS-CoV-2 inhibitor approved for outpatient use against COVID-19[13]. Intermediate results of the early months of a large efficacy trial revealed an encouraging 50% reduction in hospitalizations in the treatment group, but later analysis of the full dataset showed only a 30% lower hospitalization rate overall[14]. Based on the geographical location of trial participants and VOC prevalence in the earlier versus later phase of the trial, an

[1]Center for Translational Antiviral Research, Institute for Biomedical Sciences, Georgia State University, Atlanta, GA 30303, USA. [2]Virology Division, Department of Laboratory Medicine, University of Washington, Seattle, WA 98185, USA. [3]Emory Institute for Drug Development, Emory University, Atlanta, GA 30329, USA. [4]Department of Pathology, College of Veterinary Medicine, University of Georgia, Athens, GA 30602, USA. [5]These authors contributed equally: Carolin M. Lieber, Robert M. Cox. ✉e-mail: rplemper@gsu.edu

advisory board to the FDA considered the lower efficacy of molnupiravir against VOC delta as a possible explanation for the mixed results[15]. However, VOC delta was efficiently inhibited by the molnupiravir parent metabolite N[4]-hydroxycytidine (NHC) in ex vivo studies[16], suggesting unchanged sensitivity to the drug.

Human airway epithelium organoids, ferrets, mice, and Syrian golden hamsters have emerged as preclinical models to assess the efficacy of anti-SARS-CoV-2 drug candidates. Ferrets recapitulate the predominant clinical presentation of SARS-CoV-2 in younger patients, characterized by a high viral load in the upper respiratory tract, strong viral shedding, and efficient airborne transmission[17–19]. By contrast, Syrian golden hamsters infected with SARS-CoV-2 develop transient pneumonia but do not recapitulate hallmark features of life-threatening severe COVID-19. Disease typically remains mild-to-asymptomatic in golden hamsters and animals fully recover within two weeks[20]. Lethal disease with severe histopathology affecting lung, liver, and kidney can be induced in transgenic K18-hACE2 mice expressing human ACE2, but organ distribution of the receptor is non-physiological, resulting in rapid neuro-invasion of the virus and the development of acute, lethal viral encephalitis[21], which is not seen in human patients. Lacking is an efficacy model that recapitulates the acute lung injury of life-threatening COVID-19, thereby offering a relevant experimental platform to test the effect of molnupiravir on mitigating lung damage caused by different VOC and explore the impact of treatment on disease outcomes.

In this work, we test molnupiravir against a panel of VOC in cultured cells, human airway epithelium organoids, ferrets, and Roborovski dwarf hamsters, since these dwarf hamsters developed acute diffuse alveolar pneumonia and succumbed to infection[22,23] when infected with original SARS-CoV-2 isolates from 2020. We show that pathogenicity in dwarf hamsters is VOC-dependent with the highest lethality in the case of delta, gamma, and omicron. Independent of VOC tested, all molnupiravir-treated dwarf hamster survived infection, meeting a primary efficacy marker. However, the treatment effect size is VOC-specific and in animals infected with VOC omicron, virus load reduction was significant in males, but not females.

## Results

For quantitative comparison of molnupiravir activity against VOC alpha, beta, gamma, delta, and omicron in cultured cells, we determined half-maximal antiviral concentrations ($EC_{50}s$) on VeroE6 cells stably expressing TMPRSS2 (Vero-TMPRSS2) (Fig. 1a). The molnupiravir parent compound NHC was used in all ex vivo experiments. Under these experimental conditions, inhibitory values were similar for the original SARS-CoV-2 WA1 isolate and all VOC tested, ranging from 0.19 (gamma) to 0.62 (beta) μM.

### Consistent activity of molnupiravir against VOC in human airway organoids

To validate activity in disease-relevant primary human tissues, we added at the time of infection NHC at 10 μM, equivalent to NHC plasma concentrations reached in human patients[24], to the basolateral chamber of well-differentiated primary human airway epithelium (HAE) cultures grown at the air-liquid interface. After apical infection with the panel of five VOC, we titered apically shed progeny virions after 1, 2, and 3 days. The treatment instantly suppressed virus replication in the case of VOC alpha, beta, and delta, and within 2 days reduced the progeny load of all VOC to the level of detection (Fig. 1b). Parallel assessment of transepithelial electrical resistance (TEER) as an indicator of an intact tight junction network demonstrated that the drug preserved the integrity of the infected organoids (Supplementary Fig. 1). vehicle-treated controls showed significant reductions in TEER after infection, reflecting the breakdown of epithelium organization. Confocal microscopy confirmed that 10 μM basolateral NHC suppressed viral replication, whereas abundant viral antigen (SARS-CoV-2

nucleocapsid and spike protein) was detected in vehicle-treated epithelia (Fig. 1c, Supplementary Figs. 2, 3).

### Efficacy of molnupiravir against VOC upper respiratory disease and transmission in ferrets

To assess in vivo efficacy of molnupiravir against VOC, we infected ferrets intranasally with $1 \times 10^5$ pfu of VOC alpha, beta, gamma, delta, and omicron and monitored virus replication in the upper respiratory tract. All treated animals were dosed orally at 5 mg/kg b.i.d., starting 12 hours after infection (Fig. 2a) when shed SARS-CoV-2 becomes first detectable in ferret nasal lavages[18,19]. Lavage titers were determined in 12-hour intervals for the first 48 hours after infection, and once daily thereafter. Titers of shed VOC alpha, gamma, delta, and omicron peaked 1–2 days after infection at approximately $10^3$ to $10^4$ pfu per ml in vehicle-treated animals (Fig. 2b). Treatment with molnupiravir reduced shed progeny titers of all VOC to detection level within 12 hours. Consistent with previous experience with the ferret model[18,19], infected animals developed no clinical signs. VOC beta did not establish a productive infection and was eliminated from further ferret experiments (Supplementary Fig. 3).

To explore the impact of treatment on transmission, we co-housed infected and treated source animals with uninfected and untreated sentinels for 48 hours, starting 42 hours after initiation of treatment (Fig. 2c). VOC delta was excluded from transmission studies, since we had not detected any infectious particles in nasal lavages of molnupiravir-treated animals at any time and nasal turbinates of treated animals extracted four days after infection did not contain any infectious particles (Supplementary Fig. 5). We tested VOC alpha in an independent transmission study, but included transmission arms in the efficacy studies with VOC gamma and omicron (Fig. 2b) to reduce overall animal numbers. After separation of source and contact animals on day 4 after infection, untreated sentinels were monitored for an additional four days for shed virions and viral RNA in nasal lavages, and then terminal titers in nasal turbinates were determined.

VOC alpha and gamma spread efficiently from vehicle-treated source animals to the sentinels (Fig. 2d, e), whereas VOC omicron did not transmit (Fig. 2f). RNA of VOC alpha and gamma became first detectable in nasal lavages of the contacts within 12 hours after initiation of co-housing, shed infectious particles emerged after 12 (VOC alpha) to 36 (VOC gamma) hours (Fig. 2e, f), and infectious alpha and gamma, but not omicron, particles were detectable in nasal turbinates extracted from sentinels of vehicle-treated source animals at study end (Supplementary Fig. 6). Treatment with molnupiravir fully suppressed transmission. No infectious particles or viral RNA were detectable in lavages of untreated sentinels, and nasal turbinates of these animals were virus and viral RNA-free at terminal assessment (Fig. 2e, f; Supplementary Fig. 6). These data demonstrate that oral molnupiravir is highly effective in controlling replication of all VOC in the ferret upper respiratory tract, significantly reducing shed virus titers and rapidly suppressing the spread of transmission-competent VOC to untreated naïve contacts.

### Different degrees of acute lung pathogenesis of VOC in Roborovski dwarf hamsters

To explore VOC pathogenicity in a candidate model of lethal COVID-19, we infected Roborovski dwarf hamsters intranasally with $1 \times 10^5$ pfu each of the original SARS-CoV-2 WA1 isolate or VOC alpha, beta, gamma, and omicron, or $3 \times 10^4$ pfu of VOC delta (Fig. 3a) and monitored clinical signs and survival. Dwarf hamsters showed a rapid decline characterized by ruffled fur, lethargy, and dyspnea within two days of infection, which was accompanied by hypothermia and moderate to substantial loss of body weight (Fig. 3b, c). Severity and time to onset of clinical signs varied among—in order of increasing pathogenicity—VOC alpha, beta, omicron, gamma, and delta. Pathogenesis of VOC gamma was comparable to that of the original WA1 isolate. The

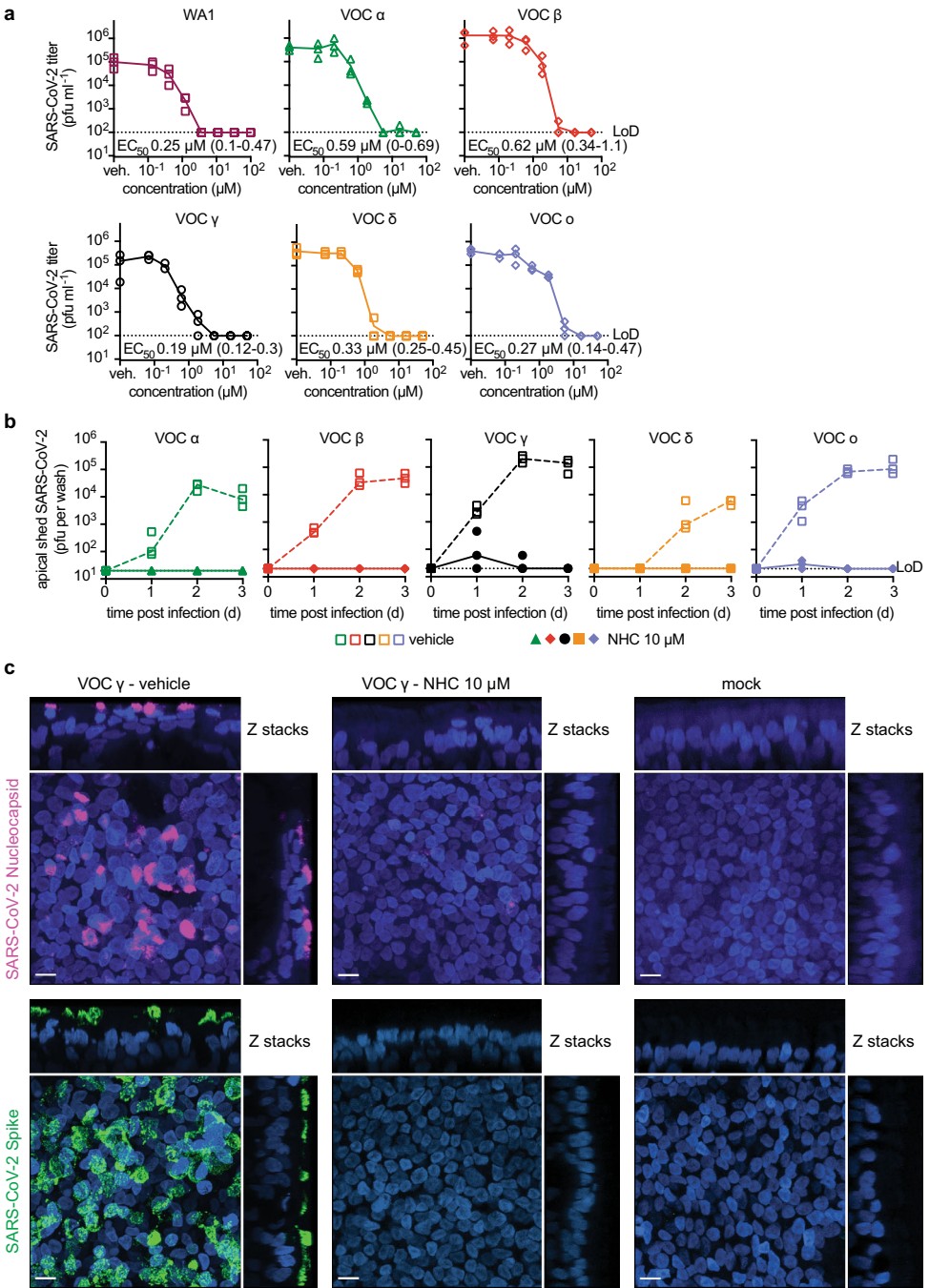

**Fig. 1 | Efficacy of NHC against VOC in cultured cells and primary HAEs. a** NHC dose-response assays against SARS-CoV-2 WA1 and VOC alpha, beta, gamma, delta, and omicron on VeroE6-TMPRSS2 cells. $EC_{50}$ values from non-linear regression modeling are shown with 95% confidence intervals in parenthesis. **b** Activity of 10 μM basolateral NHC administered against VOC as in **a** on well-differentiated primary HAE organoids. Apically shed virus was harvested every 24 hours during a 3-day period after infection. **a**, **b** represent independent biological repeats, lines intersect group means. LoD limit of detection. **c** Confocal microscopy of primary HAE organoids infected with SARS-CoV-2 VOC gamma. Basolateral NHC suppresses virus replication at 10 μM. SARS-CoV-2 nucleocapsid (pink; α-SARS-CoV-2 nucleocapsid), SARS-CoV-2 spike (green; α-SARS-CoV-2 S) and nuclei (DAPI) were detected. Z stacks are shown, scale bars 10 μm. Per condition and viral target, two independent transwells were processed, one stained for SARS-CoV-2 spike and one for SARS-CoV-2 nucleocapsid; representative fields of view are shown. Source data are provided as a Source Data file.

median survival of infected animals was shortened to 5.75 (WA1), 6 (omicron), 3.5 (beta, gamma), and 3 (delta) days after infection (Fig. 3d; Table 1). Approximately 80% of animals inoculated with VOC alpha survived the infection, making alpha the least lethal of the VOC tested.

We intended to determine viral tissue tropism and gross lung pathology three days after infection of subsets of dwarf hamsters with WA1 or the VOC associated with the overall highest mortality rate, gamma, delta, and omicron. However, animals infected with

delta succumbed to infection before the predefined endpoint, preventing analysis. Animals infected with WA1, gamma, or omicron consistently showed the highest viral RNA load in the lung, followed by the small and large intestine and spleen (Fig. 3e). Macroscopic analysis of extracted lungs showed substantial gross tissue damage involving large surface areas (Fig. 3f, Supplementary Fig. 7) and titration of lung homogenates confirmed fulminant viral pneumonia with median virus load of $1 \times 10^5$ pfu/g lung tissue or greater (Fig. 3g).

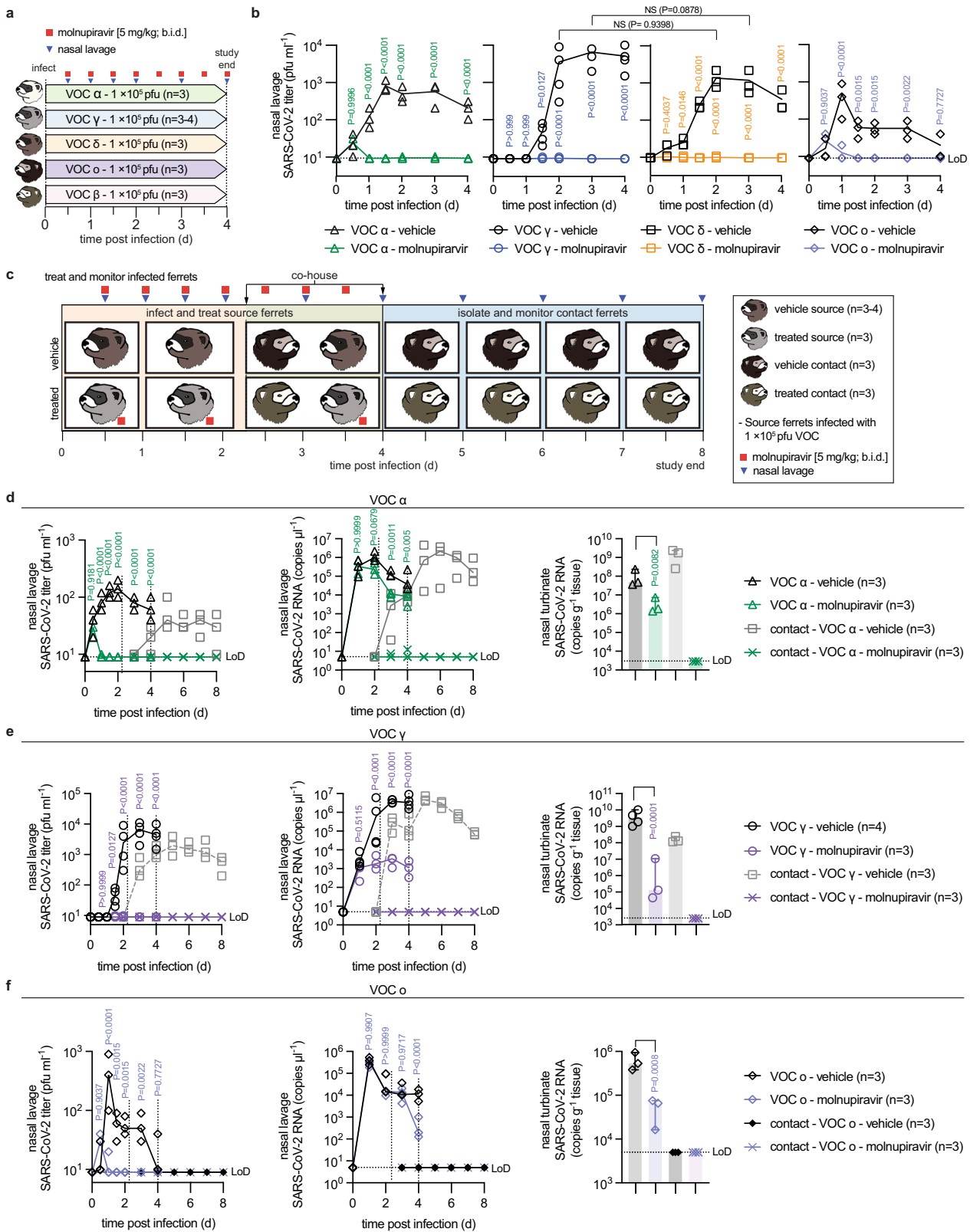

A follow-up study with VOC delta at reduced inoculum amount ($1 \times 10^4$ pfu intranasally) and shortened time to analysis (Fig. 3h) demonstrated the onset of lung tissue damage (Fig. 3i, Supplementary Fig. 8) and presence of high lung virus loads (Fig. 3j) only 12 hours after infection. Comparison analysis of VOC gamma 24 hours after infection confirmed that rapid viral invasion of the dwarf hamsters was not limited to VOC delta.

## Effect of VOC type and biological sex on prevention of lethal viral pneumonia

VOC gamma, delta, and omicron were selected for an efficacy study assessing mitigation of viral pneumonia and acute lung injury with molnupiravir. Animals were inoculated with $1 \times 10^4$ pfu intranasally to prevent premature death, followed by initiation of treatment (250 mg/kg orally b.i.d.) 12 hours after infection (Fig. 4a), when lung

**Fig. 2 | Therapeutic molnupiravir is efficacious against a panel of VOC in ferrets.** **a** Treatment and monitoring schematic. Ferrets (*n* = 6 total per VOC tested) were infected intranasally with $1 \times 10^5$ pfu of VOC and treated orally with vehicle or molnupiravir (5 mg/kg b.i.d.), starting 12 hours after infection. **b** Infectious SARS-CoV-2 titers in nasal lavages of vehicle or molnupiravir-treated animals. **c** Schematic of transmission studies with VOC alpha, gamma, and omicron. For VOC alpha, fresh source ferrets (*n* = 6) were used, for VOC gamma and omicron, transmission arms were added to **b**. Source animals were co-housed at a 1:1-ratio with uninfected and untreated contacts starting 42 hours after the beginning of treatment. Nasal lavages were obtained twice daily in the first 48 hours after infection, thereafter once daily. Nasal turbinates were extracted from source animals 4 days after infection and untreated sentinels continued for an additional 4 days. **d**–**f** Transmission study with VOC alpha (**d**), gamma (**e**), and omicron (**f**). Shown are infectious titers in nasal lavages (left), viral RNA copies in the lavages (center), and nasal turbinate titers (right). Symbols in **b**, **d**–**f** represent independent biological repeats (virus load of individual animals), lines (**b**, **d**–**f**) intersect group medians, and columns (**c**, **e**–**g**) show group medians ±95% confidence intervals. Statistical analysis with one-way (**d**–**f** turbinate titers) or two-way (**b**, **d**–**f** lavage titers) ANOVA with Tukey's (**d**–**f** turbinate titers) or Sidak's (**b**, **d**–**f** lavage titers) posthoc multiple comparison tests; *P* values are shown, NS not significant, LoD limit of detection. Source data are provided as a Source Data file.

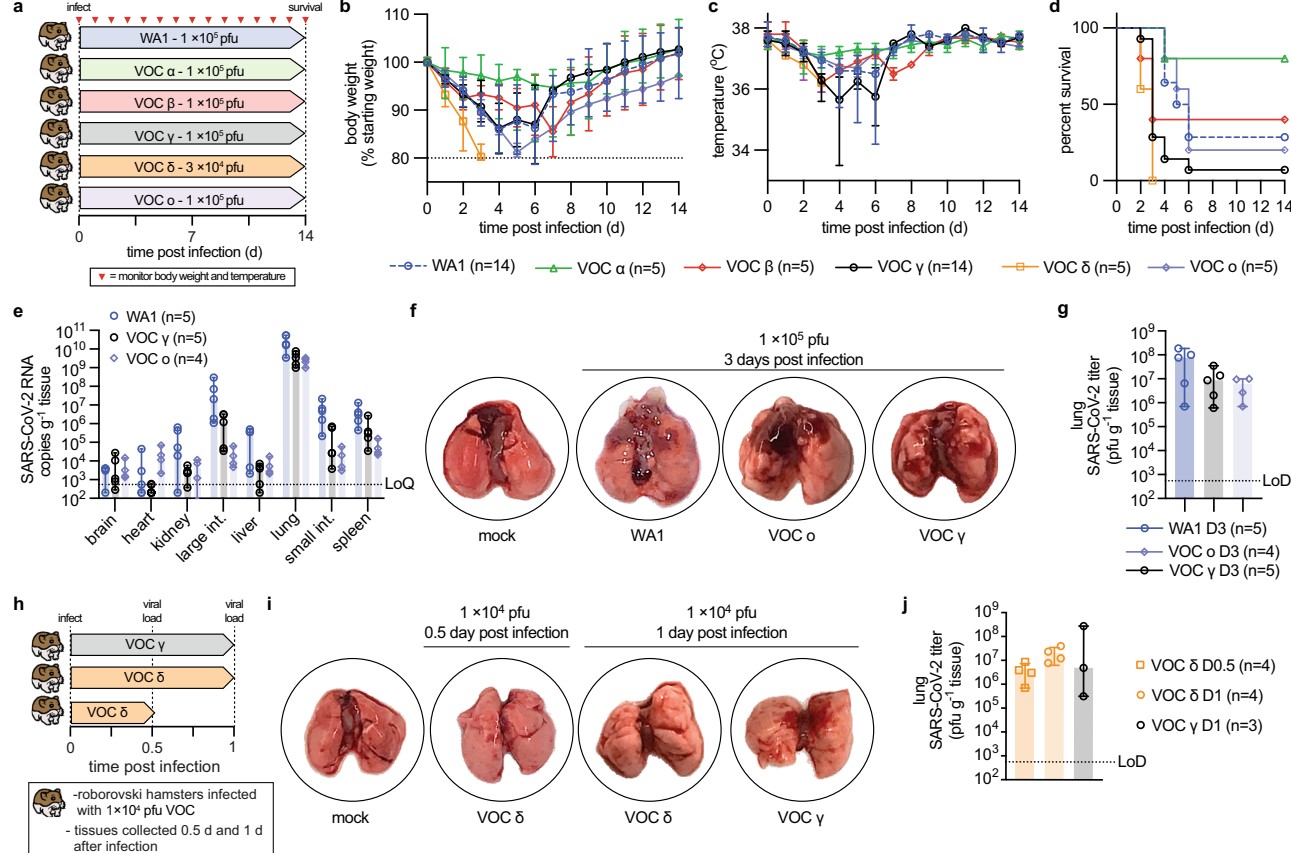

**Fig. 3 | VOC pathogenesis in Roborovski dwarf hamsters. a** Schematic of the dwarf hamster pathogenesis study. Animals were monitored for up to 14 days after intranasal infection with $1 \times 10^5$ pfu each (VOC delta $3 \times 10^4$ pfu per animal). Clinical signs were assessed once daily (red triangles). **b**, **c** Body weight (b) and temperature (c) of infected dwarf hamsters. **d** Survival curves of infected dwarf hamsters from **a**. **e** Viral RNA copies in select organs extracted from infected dwarf hamsters 3 days after infection. **f** Images of lungs from dwarf hamsters mock infected or inoculated with $1 \times 10^5$ pfu of WA1, VOC omicron, or VOC gamma 3 days after infection. **g** Infectious titers from the lungs of dwarf hamsters shown in **f**. **h** Schematic of the dwarf hamster pathogenesis study utilizing $1 \times 10^4$ pfu of VOC delta and gamma. **i** Images of lungs extracted from dwarf hamsters mock infected or inoculated with $1 \times 10^4$ pfu of VOC omicron or VOC gamma 0.5 (delta) and 1 day (delta and gamma) after infection. **j** Infectious titers from the lungs of dwarf hamsters shown in **i**. Symbols represent independent biological repeats (**e**, **g**, **j**), lines intersect group medians (**b**, **c**), columns show group medians (**e**, **g**, **j**), and error bars represent 95% confidence intervals. LoD limit of detection, LoQ limit of quantitation. Source data are provided as a Source Data file.

virus load was high and first lung lesions became detectable (Fig. 3i, j). The higher molnupiravir dose compared to that administered to ferrets was used to compensate for the high metabolic activity of the dwarf hamsters[25] and is consistent with the dose level administered to other rodent species[26,27]. Oral molnupiravir alleviated clinical signs (Supplementary Fig. 9) and ensured complete survival of all treated animals independent of VOC type, whereas approximately 50% (gamma) and 90% (delta, omicron) of animals in the vehicle groups succumbed to the infection within 2 to 7 days (Fig. 4b). Lung virus load assessed three days after infection was consistently high (approximately $10^7$–$10^8$ pfu/g lung tissue) in all vehicle-treated groups (Fig. 4c). Treatment significantly lowered lung titers

independent of VOC type, but effect size varied greatly from approximately 1 (delta) to over 4 (gamma) orders of magnitude. Although likewise statistically significant compared to vehicle-treated animals, the impact of molnupiravir on omicron lung load showed major variation between individual animals, creating low, high, and super responder groups characterized by lung titer reductions of ~1, 4, and >5 orders of magnitude, respectively (Fig. 4c). Infectious titers in the lung were closely mirrored by viral RNA copies present in lung and tracheas (Supplementary Fig. 10).

Having powered this study with approximately equal numbers of male and female animals in each group, we queried the dataset for a possible impact of biological sex on outcome (Fig. 4d). No significant

**Table 1 | Analysis of dwarf hamster survival after infection with different VOC**

| ID | Median survival (d) | Survival (%) | Animal number (n) | Comparison | P value |
|---|---|---|---|---|---|
| WA1 | 5.75 | 28.571 | 14 | – | n.a. |
| | | | | VOC α | 0.0873 |
| | | | | VOC β | 0.6119 |
| | | | | VOC γ | 0.0031 |
| | | | | VOC δ | <0.0001 |
| | | | | VOC o | 0.9988 |
| VOC α | n.d. | 80 | 5 | – | n.a. |
| | | | | VOC β | 0.1591 |
| | | | | VOC γ | 0.0032 |
| | | | | VOC δ | 0.0031 |
| | | | | VOC o | 0.1177 |
| VOC β | 3 | 40 | 5 | – | n.a. |
| | | | | VOC γ | 0.3278 |
| | | | | VOC δ | 0.1729 |
| | | | | VOC o | 0.8741 |
| VOC γ | 3 | 7.143 | 14 | – | n.a. |
| | | | | VOC δ | 0.0505 |
| | | | | VOC o | 0.0555 |
| VOC δ | 3 | 0 | 5 | – | n.a. |
| | | | | VOC o | 0.0031 |
| VOC o | 6 | 20 | 5 | – | n.a. |

Significance of differences in survival curves was determined with a two-sided Log-rank (Mantel–Cox) test. Source data are provided as a Source Data file.

differences in lung virus load between males and females were detected in any of the three vehicle groups, which was consistent with the absence of a correlation between biological sex and probability of survival of the vehicle-treated animals in our survival study (Supplementary Fig. 11). Comparison of lung virus burden of males and females in the molnupiravir-treated groups revealed no statistically significant differences in effect size in animals infected with VOC gamma or delta, but biological sex had a statistically significant influence on molnupiravir benefit of animals infected with omicron (Fig. 4d, Table 2). Whereas lung titer reductions in treated males were highly significant compared to vehicle-treated males, females, or all vehicle-treated animals combined, no significant change in lung virus load against any of these vehicle groups was detected in females treated with molnupiravir.

## Pharmacokinetic (PK) properties of molnupiravir in dwarf hamsters

To explore whether in the dwarf hamster model PK properties of molnupiravir are unexpectedly affected by biological sex of the animals, we determined plasma profiles and tissue distribution after a single oral molnupiravir dose of 250 mg/kg body weight, administered to groups of males and females. As observed in other species[28], the ester prodrug was rapidly hydrolyzed at intestinal absorption, leaving essentially only the molnupiravir parent NHC detectable in plasma samples (Fig. 5a). No major differences in PK parameters were observed between male and female dwarf hamsters (Fig. 5b, Supplementary Table 2). Tissue exposure of the corresponding bioactive anabolite NHC triphosphate (NHC-TP) was highest in spleen and lung, sustained in respiratory tissues at levels exceeding 10 nmol NHC-TP/g tissue over a ≥8-hour period after dosing, and consistent between male and female animals (Fig. 5c).

## VOC-specific adaptation of SARS-CoV-2 in dwarf hamsters

In earlier studies with SARS-CoV-2 in the ferret model[18], we noted the rapid appearance of characteristic host adaptation mutations such as an L260F substitution in nsp6 and a Y453F mutation in spike[29] in virus populations extracted from ferret nasal turbinates. To probe for possible virus adaptations to the dwarf hamsters, we sequenced whole genomes of virus populations recovered from the different vehicle or molnupiravir treatment groups. No dwarf hamster-typical mutations were detected that were dominant across all VOC populations, but we detected several VOC type-specific substitutions with >20% allele frequency compared to the respective virus inoculum. Irrespective of treatment status, a spike D142G substitution was present in nearly all VOC delta populations, but no dominating mutations emerged in spike proteins of the different VOC gamma and omicron dwarf hamster reisolates (Supplementary Table 1). All recovered VOC gamma populations harbored an nsp6 V181F substitution and all VOC omicron populations contained the nsp6 L260F mutation that was originally considered to be characteristic for adaptation to weasels[18]. However, none of the recovered VOC delta populations contained substitutions in nsp6. We found isolated additional substitutions in some virus populations recovered from individual animals in the respective infection and treatment groups (Supplementary Table 1), but detected no correlation to relative viral fitness in vehicle-treated dwarf hamsters or link to molnupiravir treatment success.

## Molnupiravir-mediated mitigation of lung histopathology

Macroscopic assessment of the lungs extracted three days after infection revealed severe tissue damage with large lesions covering ~30% (omicron) to 50% (gamma, delta) of the lung surface area of vehicle-treated animals (Fig. 6a, Supplementary Fig. 12). Molnupiravir treatment significantly reduced macroscopic tissue damage independent of VOC type (Fig. 6a, Supplementary Fig. 13). Histological examination of lungs extracted from animals infected with VOC gamma and delta revealed markers of severe viral infection in vehicle-treated animals, including perivascular cuffing, alveolitis, hyalinization of blood vessels, interstitial pneumonia, and leucocyte infiltration (Fig. 6b, Supplementary Fig. 14). One of the VOC gamma-infected animals developed pronounced peribronchiolar metaplasia. Due to the high lethality of VOC delta, only one animal of the vehicle group reached the predefined endpoint for tissue harvest in this study, whereas the others died prematurely and could not be examined. Molnupiravir alleviated histopathology associated with either VOC, decreasing immune cell infiltration and reducing signs of inflammation. Greater residual damage was detected in treated animals infected with VOC delta compared to gamma, which was consistent with the significantly greater molnupiravir-mediated reduction in gamma lung load detected in the efficacy study (Fig. 4c).

Immunohistochemistry analysis of the lung sections identified abundant viral antigen in animals of the vehicle-treated groups (Fig. 6c). Lung sections of molnupiravir-treated animals returned variable results depending on VOC, ranging from strong staining after infection with delta to complete absence of viral antigen after VOC gamma. This differential staining intensity recapitulated the differences observed in lung virus load and viral RNA copies after treatment of animals infected with VOC delta versus gamma (Fig. 4c, Supplementary Fig. 9). When we examined lung sections of molnupiravir-treated animals and surviving members of the vehicle groups two weeks after infection, no viral antigen was detectable and only minor signs of infection were visible (Supplementary Fig. 15), indicating that histopathological damage was transient in survivors.

These results demonstrate that acute lung injury occurs rapidly in the dwarf hamster model. Molnupiravir consistently improves clinical signs and overall disease outcomes independent of infecting VOC. However, the degree of lung virus load reduction is greatly affected by

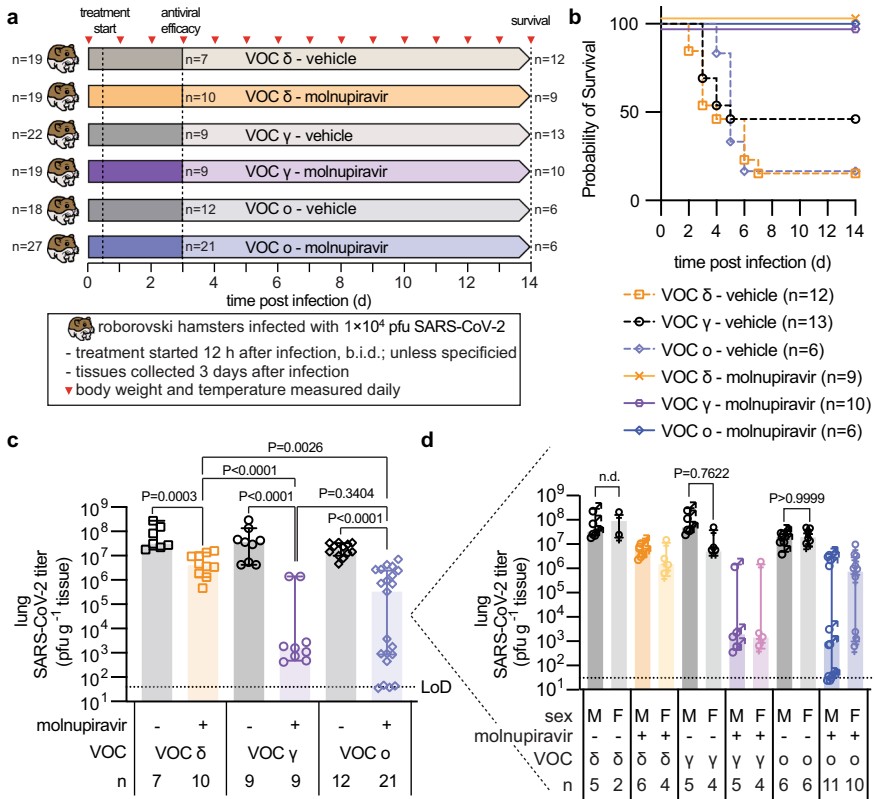

**Fig. 4 | Effects of therapeutic molnupiravir on SARS-CoV-2 lung virus load.**
**a** Schematic of the dwarf hamster efficacy study. Animals were monitored for up to 14 days after intranasal infection with $1 \times 10^4$ pfu each. Clinical signs were assessed once daily (red triangles). Groups of dwarf hamsters were euthanized 3 days after infection to assess viral load and perform histological examinations. **b** Survival curves of infected dwarf hamsters from **a**. **c** Infectious titers from the lungs of dwarf hamsters 3 days after infection as shown in **a**. **d** Role of biological sex on antiviral efficacy shown in **c**. Symbols represent independent biological repeats, columns show group medians (**c**, **d**), and error bars represent 95% confidence intervals. Significance was determined using unpaired two-tailed *t* tests (**c**) or two-way ANOVA with Sidak's posthoc multiple comparison tests without further adjustments (**d**); *P* values are shown. Source data are provided as a Source Data file.

VOC type and, in the case of infection with VOC omicron, influenced by the biological sex of the treated animal.

## Discussion

This study demonstrates efficacy of molnupiravir against relevant SARS-CoV-2 VOC alpha, gamma, delta, and omicron in human airway organoids, the ferret transmission model, and the Roborovski dwarf hamster model of life-threatening viral pneumonia. We show that the dwarf hamsters provide a robust experimental system to explore degrees of pathogenicity of different VOC[30–33]. The implications of these efficacy models for the anticipated pharmacological impact of molnupiravir are four-fold:

i. Direct antiviral potency of the molnupiravir parent NHC against all VOC including delta and omicron was virtually identical in cultured cells and human organoids, and closely resembled that reported for the original WA1 isolate[18]. We consider this outcome unsurprising since extensive past attempts to induce resistance to molnupiravir through adaptation of different viral targets had remained unsuccessful[28,34,35]. Based on a high genetic barrier preventing viral escape and the broad antiviral indication spectrum of molnupiravir[18,34,36], there is no molecular rationale why genetically closely related SARS-CoV-2 VOC that has no history of prior molnupiravir exposure should be inherently less sensitive to the drug.

ii. Treatment with molnupiravir significantly reduced VOC shedding from the upper respiratory tract of ferrets, shortening the time in which a host is infectious. Accordingly, molnupiravir suppressed transmission to untreated sentinels. No VOC-specific differences in drug efficacy were apparent. However, all VOC, but gamma,

also grew to lower titers in the ferret upper respiratory tract than WA1[18] and omicron did not transmit at all. By contrast, original SARS-CoV-2 isolates spread readily between ferrets through contact and airborne transmission[37] and direct and reverse zoonotic transmission in the field have been reported for closely related minks[38]. Presumably reflecting increasing SARS-CoV-2 adaptation to the human host, this reduced ferret permissiveness for SARS-CoV-2 VOC interfered with a meaningful comparison analysis of treatment efficacy.

iii. It was speculated that the variable clinical performance of molnupiravir in the interim versus complete trial analysis could be attributable to lower effectiveness of the drug against VOC delta[15]. Delta indeed surged only when the trial was underway and a large subgroup of trial participants was enrolled from the South American region[14], in which gamma happened to be dominant until Fall 2021. By revealing a strong correlation between effect size of molnupiravir efficacy and VOC type under controlled experimental conditions, the dwarf hamster model suggests VOC-dependent therapeutic benefit as a tangible explanation for the clinical conundrum. At present we cannot conclusively address, however, whether the difference in virus load reduction between VOC gamma and delta in dwarf hamsters recapitulates the variable clinical success of molnupiravir because of a common mechanism or due to a phenocopy effect. Although VOC delta and gamma replicated to similar lung titers in vehicle-treated animals, delta was associated with the shortest time-to-death. Individual VOC could indirectly modulate molnupiravir PK properties differentially by spreading to organs other than lung with distinct kinetics. However, our analysis of viral organ distribution in the

**Table 2 | Analysis of biological sex as a variable in molnupiravir efficacy in dwarf hamsters**

| | | | VOC delta | |
|---|---|---|---|---|
| | | | molnupiravir | |
| | | all | male | female |
| molnupiravir | all | n.a. | 0.9992 | 0.987 |
| | male | 0.9992 | n.a. | 0.7337 |
| | female | 0.987 | 0.7337 | n.a. |
| vehicle | all | 0.0003 | 0.0126 | 0.0003 |
| | male | 0.0011 | 0.0272 | 0.0007 |
| | female | n.d. | n.d. | n.d. |

| | | | VOC gamma | |
|---|---|---|---|---|
| | | | molnupiravir | |
| | | all | male | female |
| molnupiravir | all | n.a. | 0.9999 | 0.9999 |
| | male | 0.9999 | n.a. | 0.9999 |
| | female | 0.9999 | 0.9999 | n.a. |
| vehicle | all | 0.0001 | 0.0001 | 0.0001 |
| | male | 0.0001 | 0.0001 | 0.0001 |
| | female | 0.0006 | 0.0021 | 0.0056 |

| | | | VOC omicron | |
|---|---|---|---|---|
| | | | molnupiravir | |
| | | all | male | female |
| molnupiravir | all | n.a. | 0.9889 | 0.9857 |
| | male | 0.9889 | n.a. | 0.5376 |
| | female | 0.9857 | 0.5376 | n.a. |
| vehicle | all | 0.0002 | 0.0001 | 0.0611 |
| | male | 0.0067 | 0.0012 | 0.2249 |
| | female | 0.0063 | 0.0012 | 0.2155 |

Significance was determined using two-way ANOVA with Sidak's multiple comparisons post-hoc test without further adjustments. Numbers specify *P* values; dark grey shading, *P* values >0.05; light grey shading, *P* values <0.05. Source data are provided as a Source Data file.

dwarf hamsters revealed very low viral RNA burden in liver, the primary site of drug metabolism[39], and sustained antiviral NHC-TP exposure levels were reached in all tissues but brain. Alternatively, the rapid-onset lung histopathology seen with VOC delta may directly contribute to lower effect size of therapy.

iv. Unexpectedly, molnupiravir efficacy against VOC omicron was variable between individual dwarf hamsters. Biological sex of the animals emerged as a correlate for therapeutic benefit of molnupiravir use against omicron, with treated males faring better overall than females. By contrast, biological sex had no effect on treatment benefit when dwarf hamsters were infected with VOC gamma or delta, which matched human trial data reported for

these VOC[14]. Dwarf hamsters are outbred and animals used in this study were not raised under controlled conditions, introducing individual differences in body weight, age, microbiome, drug metabolism, and/or prior disease history as additional variables, which certainly are all equally present also in human patients. However, dwarf hamsters were randomly assigned to the different study groups and these factors, if indeed of importance, should have resulted in equal individual variation in viral load in the vehicle group or in animals infected with VOC gamma or delta. Whole genome sequence analysis of VOC omicron populations recovered from the dwarf hamsters at the end of infection revealed furthermore no correlation between potential differential VOC omicron adaptation to the dwarf hamster host and effect size of molnupiravir therapy, pointing overall to high variability of omicron disease dynamics in treated dwarf hamsters.

In the absence of controlled clinical data assessing molnupiravir efficacy against omicron, it is currently unclear to what degree the dwarf hamster-derived results extend to human therapy. Our study demonstrates, however, that pharmacological mitigation of severe COVID-19 is complex and that attempts to predict drug efficacy based on unchanged ex vivo inhibitory concentrations alone[16] may be premature. The dwarf hamster-based results illuminates that VOC-specific differences in treatment effect size may be present in vivo, alerting to the need to continuously reassess therapeutic benefit of approved antivirals for individual patient subgroups as SARS-CoV-2 evolves and potential future VOC may emerge.

## Methods
### Ethics statement
All experiments involving infectious SARS-CoV-2 were approved by the Georgia State Institutional Biosafety Committee under protocol B20016 and performed in BSL-3/ABSL-3 facilities at the Georgia State University.

All animal studies were performed in compliance with the Guide for the Care and Use of Laboratory Animals of the National Institutes of Health and the Animal Welfare Act Code of Federal Regulations. Experiments with SARS-CoV-2 involving ferrets and dwarf hamsters were approved by the Georgia State Institutional Animal Care and Use Committee under protocols A20031 and A21019, respectively.

### Study design
This study used female ferrets (*Mustela putorius furo*) family mustelids, genus mustela, 6–10 months of age, and male and female Roborovski dwarf hamsters (*Phodopus roborovskii*), family cricetidae, genus phodopus, 3-10 months of age as in vivo models to assess the therapeutic efficacy of orally administered molnupiravir against infections with different SARS-CoV-2 VOC. Ferrets were used to examine the effect of molnupiravir on upper respiratory infection and transmission. Roborovski dwarf hamsters were used to study the effects of molnupiravir treatment on severe disease associated with lower respiratory tract infection and acute lung injury. VOC were administered to animals through intranasal inoculation. For ferrets, upper respiratory tract viral titers were assessed routinely through nasal lavages and upper respiratory tract tissues. For dwarf hamsters, animals were monitored twice daily for clinical signs, and viral loads were determined in respiratory tract tissues at endpoint. Virus loads were determined by plaque assays and RT-qPCR quantitation.

### Cells
African green monkey kidney cells VeroE6 (ATCC CRL-1586™), Calu-3 (ATCC HB-55™), and VeroE6-TMPRSS2 (BPS Bioscience #78081) were cultivated at 37 °C with 5% $CO_2$ in Dulbecco's Modified Eagle's Medium (DMEM) supplemented with 7.5% heat-inactivated fetal bovine serum (FBS). Normal human bronchial/tracheal epithelial cells (NHBE) (Lonza

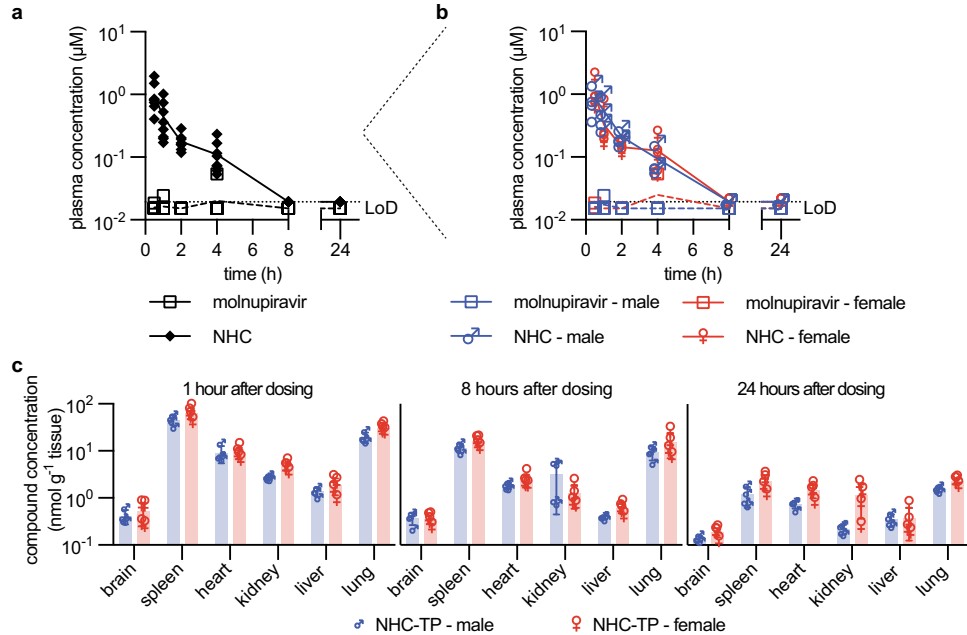

**Fig. 5 | Single oral dose PK properties of molnupiravir in Roborovski dwarf hamsters.** Male and female dwarf hamsters were gavaged with molnupiravir at (250 mg/kg body weight). **a** Plasma concentrations of molnupiravir and NHC, determined over a 24-hour period after dosing. **b** Plasma concentrations from **a**, showing male and female subgroups; LoD level of detection (0.0193 μM for NHC (shown); 0.0152 μM for molnupiravir). **c** Tissue distribution of NHC-TP 1-, 8-, and 24-hours after dosing in male and female dwarf hamsters. Symbols show results for individual animals (**a**–**c**), columns represent group means (**c**); *n* = 4 each per biological sex and time point (**a**–**c**). Source data are provided as a Source Data file.

Bioscience, cat# CC-2540S, lot# 0000646466, passage 3, donor M4) from a 38-year-old male were expanded in PneumaCult-Ex Plus (Stemcell Technologies cat# 05040) and differentiated in PneumaCult-ALI (Stemcell Technologies cat# 05001) for 8 weeks following the manufacturer's instructions. To protect the privacy of donors and tissue suppliers, Lonza Bioscience does not provide copies of donor records or tissue source agreements to customers. Lonza holds donor consent and legal authorizations that give permission for all research use. These consent and authorization documents do not identify specific types of research testing. If used for research purposes only, the donor consent applies. All cells were authenticated and checked for mycoplasma prior to use.

## Antibodies

SARS-CoV-2 N and goblet cells were co-stained using rabbit anti-SARS-CoV-2 Nucleocapsid monoclonal antibody (HL453) (Invitrogen, cat# MA5-36272; validation datasheet at https://www.thermofisher.com/antibody/product/SARS-CoV-2-Nucleocapsid-Antibody-clone-HL453-Monoclonal/MA5-36272) (1:100 dilution) and mouse anti-MUC5AC (45M1) (ThermoFisher, cat# MA5-12175; validation datasheet at https://www.thermofisher.com/antibody/product/MUC5AC-Antibody-clone-45M1-Monoclonal/MA5-12175) (1:200 dilution) as primary antibodies, respectively. Donkey anti-goat (Alexa Fluor® 568 (ThermoFisher Scientific, cat# A-11057; validation datasheet at https://www.thermofisher.com/antibody/product/Donkey-anti-Goat-IgG-H-L-Cross-Adsorbed-Secondary-Antibody-Polyclonal/A-11057) and rabbit anti-mouse IgG (H + L) cross-adsorbed secondary antibody (Alexa Fluor® 488) (ThermoFisher Scientific, cat# A-11059; validation datasheet at https://www.thermofisher.com/antibody/product/Rabbit-anti-Mouse-IgG-H-L-Cross-Adsorbed-Secondary-Antibody-Polyclonal/A-11059) (1:500 dilution) were used as secondary antibodies. For staining of SARS-CoV-2 S, mouse anti-SARS-CoV-1 and 2 Spike protein monoclonal antibody (1A9) (Abcam, cat# ab273433; validation datasheet at https://www.abcam.com/sars-spike-glycoprotein-antibody-1a9-ab273433.html) (1:200 dilution) and goat anti-mouse IgG (H + L) highly cross-adsorbed secondary antibody (Alexa Fluor® 488

(Invitrogen, cat# A-11029; validation datasheet at https://www.thermofisher.com/antibody/product/Goat-anti-Mouse-IgG-H-L-Highly-Cross-Adsorbed-Secondary-Antibody-Polyclonal/A-11029) (1:500 dilution), were used as primary and secondary antibodies, respectively. For staining of ciliated cells, rabbit anti-beta IV tubulin recombinant antibody conjugated with Alexa Fluor® 647 (EPR16775) (Abcam, cat# ab204034; validation datasheet at https://www.abcam.com/alexa-fluor-647-beta-iv-tubulin-antibody-epr16775-ab204034.html) (1:100 dilution) was used.

## Viruses

The following SARS-CoV-2 isolates were obtained from BEI resources, amplified on Calu-3 cells. SARS-CoV-2 (WA1; lineage A, isolate USA-WA1/2020, BEI cat# NR-52281), VOC alpha (lineage B.1.1.7, isolate USA/CA_CDC_5574/2020, BEI cat# NR-54011), VOC beta (lineage B.1.351, isolate hCoV-19/South Africa/KRISP-K005325/2020, BEI cat# NR-54009), and VOC gamma (lineage P.1., isolate hCoV-19/Japan/TY7-503/2021 (Brazil P.1), BEI cat# NR-54982). VOC delta (lineage B.1.617.2, clinical isolate #2333067) and VOC omicron (lineage B.1.1.529, WA-UW-21120120771) were obtained from the Northwestern Reference laboratory and amplified on Calu-3 cells. All viruses were authenticated by whole genome next-generation sequencing prior to use.

## Virus yield reduction

12-well plates were seeded with $2 \times 10^5$ cells per well the day before infection. Each isolate was diluted in DMEM to achieve a multiplicity of infection of 0.1 pfu/cell, adsorbed on cells for 1 hour at 37 °C followings which the inoculum was removed and replaced with DMEM with 2% heat-inactivated FBS. The media contained additionally 0.1% DMSO (vehicle) and the indicated concentration of NHC (EIDD-1931). After 48 hours at 37 °C, the cell supernatant was harvested aliquoted, and frozen at −80 °C before titration by standard plaque assay. Log viral titers were normalized using the average top plateau of viral titers to define 100% and were analyzed through 4-parameter variable slope non-linear regression modeling slope to determine $EC_{50}$ and 95% confidence intervals (Prism; GraphPad).

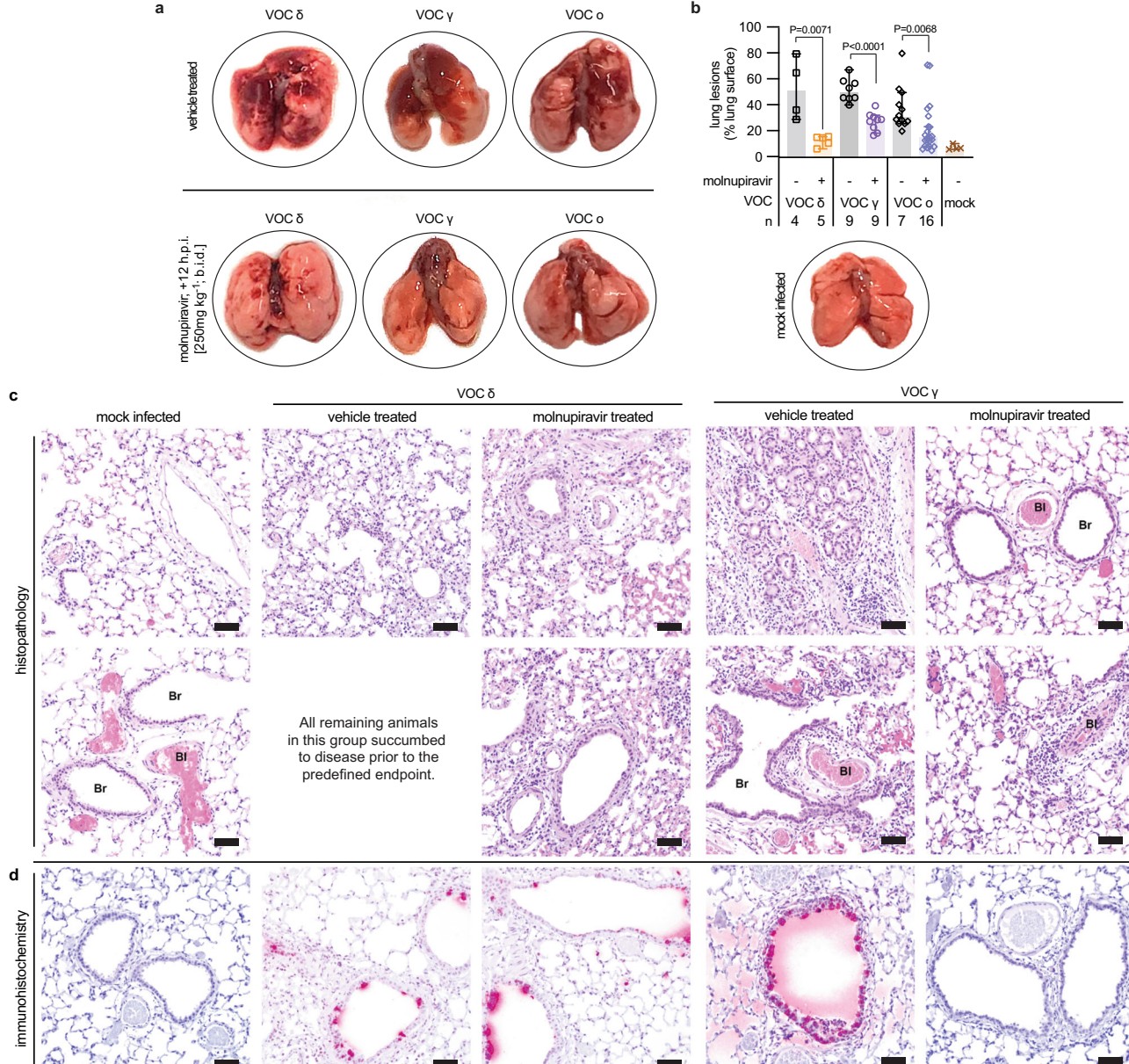

**Fig. 6 | Molnupiravir mitigates lung histopathology in VOC-infected dwarf hamsters. a** Images of lungs from dwarf hamsters mock infected or inoculated with $1 \times 10^4$ pfu of VOC delta, gamma, or omicron, treated with vehicle or molnupiravir, and harvested 3 days after infection. **b** Quantitation of macroscopic lesions as a percent of total visible lung surface area for infected dwarf hamsters treated with vehicle or molnupiravir; *n* values are shown. Mock infected lungs (*n* = 4) were included as a reference. Symbols represent independent biological repeats, columns show group medians. Significance was determined using unpaired two-tailed *t* tests; *P* values are shown. **c, d** Histopathology (**c**) and immunohistochemistry (**d**) of dwarf hamster lung slices from groups harvested 3 days after infection. Staining with hematoxylin and eosin (**c**) or α-SARS-CoV-2 S (**d**). Br bronchioles, Bl blood vessel, scale bar 50 μm. Each image represents tissue from an individual animal, all individual animals analyzed are shown in Supplementary Fig. 14. Source data are provided as a Source Data file.

## Confocal microscopy

Eight-week differentiated HAE cells were infected (or mock-infected) with $3 \times 10^4$ pfu of SARS-CoV-2 VOC and fixed on day 3 after infection. For co-staining of SARS-CoV-2 N and goblet cells, cells were permeabilized for 30 minutes with a 200 μl apical overlay of PBS with 0.1% Triton X-100 and 3% bovine serum albumin (BSA), then the overlay was removed and cells were apically incubated with 100 μl of primary antibodies in wash buffer (PBS with 0.05% Tween20 and 0.3% BSA) for 1 hour at RT. After three 5-minute 200 μl-washes with wash buffer, cells were incubated with secondary antibodies in wash buffer for 1 hour at RT. Cells were subsequently incubated with Hoechst 34580 (1:1000 dilution in wash buffer) for 5 minutes and washed three times. Membranes were then excised and mounted using Prolong Diamond antifade mountant overnight at RT (ThermoFisher Scientific). For co-staining of SARS-CoV-2 S and ciliated cells, the same method was used with a saturation step with PBS with 0.05% Tween20 and 3% BSA in lieu of a permeabilization step. Image captures were performed with a Zeiss Axio Observer Z.1 and an LSM 800 confocal microscope with AiryScan, controlled with the Zeiss Zen 3.1 Blue software package (Windows 10). Representative pictures were taken with a ×63 Plan-Apochromat (NA: 1.40, oil) objective. Digital pictures are pseudocolored for optimal presentation. For clarity of presentation, the intensity of staining was digitally balanced in between different virus isolates with Zeiss Zen 3.1 Blue, while keeping the same processing for the conditions that are directly compared (vehicle vs treatment).

## Plaque assay

Samples were serially diluted (tenfold dilutions starting at a 1:10 initial dilution) in DMEM medium supplemented with 2% FBS containing Antibiotic-Antimycotic (Gibco). The serial dilutions were added to Vero E6 cells seeded in 12-well plates at $3 \times 10^5$ cells per well 24 hours before. The virus was allowed to adsorb for 1 hour at 37 °C. Subsequently, the inoculum was removed, and the cells were overlaid with 1.2% Avicel (FMC BioPolymer) in DMEM and incubated for 3 days at 37 °C with 5% $CO_2$. After three days, the Avicel was removed, cells were washed once with PBS, fixed with 10% neutral buffered formalin, and plaques were visualized using 1% crystal violet. For dwarf hamsters infected with VOC gamma, delta, and omicron, plaque assays were performed with VeroE6-TMPRSS2 cells.

## In vivo efficacy of molnupiravir in ferrets against VOC

Ferrets were purchased from Triple F Farms, PA. Groups of ferrets were inoculated with $1 \times 10^5$ pfu of VOC alpha or delta (1 ml; 0.5 ml per nare). Twelve hours after infection, groups of ferrets were treated twice daily (b.i.d.) with vehicle (1% methylcellulose) or molnupiravir at a dose of 5 mg/kg in 1% methylcellulose. Treatments were administered by oral gavage and continued every 12 hours until 4 days after infection. All ferrets were euthanized 4 days after infection and tissues were harvested to determine SARS-CoV-2 titers and the presence of viral RNA.

## Efficacy and contact transmission of VOC in ferrets

Two groups of six source ferrets were inoculated intranasally with $1 \times 10^5$ pfu of VOC alpha, gamma, or omicron. After 12 hours, groups of source ferrets were further divided into two groups, receiving vehicle ($n = 3$ for VOC alpha; $n = 4$ for VOC gamma) or molnupiravir treatment (5 mg/kg b.i.d.) administered by oral gavage. At 54 hours after infection, each source ferret was co-housed with uninfected and untreated contacts. Ferrets were co-housed until 96 hours after infection, when source ferrets were euthanized and contact ferrets were housed individually. All contact ferrets were monitored for 4 days after separation from source ferrets and then euthanized. Nasal lavages were performed on source ferrets twice daily until cohousing started. After cohousing started, nasal lavages were performed on all ferrets every 24 hours. Nasal turbinates were harvested for all ferrets after euthanization to determine infectious titers and the presence of viral RNA.

## In vivo pathogenicity of VOC in dwarf hamsters

Female and male dwarf hamsters (2–4 months of age) were purchased from Dierengroothandel Ron Van Der Vliet, Netherlands. The dwarf hamsters were permanently quarantine-housed under ABSL-2 conditions until study start. After a minimum resting period of 2 weeks after arrival, animals were randomly assigned to groups for individual studies, transferred into an ABSL-3 facility immediately prior to study start, and housed singly in ventilated negative-pressure cages during the studies. To establish a pathogenicity profile, dwarf hamsters were inoculated intranasally with $1 \times 10^5$ pfu in 50 µl (25 µl per nare), unless otherwise stated (VOC delta $3 \times 10^4$). The dwarf hamsters were anaesthetized with dexmedetomidine/ketamine before inoculation. Groups of dwarf hamsters were euthanized 14 days after infection and their organs were harvested to determine the presence of viral RNA in different tissues (WA1 and VOC gamma only).

## In vivo efficacy of molnupiravir in dwarf hamsters

Groups of dwarf hamsters were inoculated with $1 \times 10^4$ pfu in 50 µl (25 µl per nare). At 12 hours after infection, the dwarf hamsters were treated b.i.d. with vehicle (1% methylcellulose) or molnupiravir at a dosage of 250 mg/kg body weight, respectively. The compound was administered via oral gavage in 1% methylcellulose. After the start of treatment, b.i.d. dosing was continued until 12 days after infection. Subgroups of dwarf hamsters were euthanized 3 days after infection. All studies were terminated 14 days after infection. Organs were harvested to determine virus titers and the presence of viral RNA in different tissues. For PK studies, dwarf hamsters received a single oral dose of 250 mg/kg body weight molnupiravir. Blood was sampled at 0.5, 1, 2, 4, 8, and 24 hours after dosing from four animals per biological sex and plasma extracts prepared. Individual dwarf hamsters were sampled not more than twice. Organ samples were extracted 1, 8, and 24 hours after dosing. Molnupiravir, and NHC and its anabolites were analyzed using a qualified LC/MS/MS method[28], calculations with WinNonlin 8.3.3.33.

## Titration of SARS-CoV-2 in tissue extracts

For virus titration, the organs were weighed and homogenized in PBS. The homogenates were centrifuged at $2000 \times g$ for 5 minutes at 4 °C. The clarified supernatants were harvested, frozen, and used in subsequent plaque assays. For detection of viral RNA, the harvested organs were stored in RNAlater at −80 °C. The tissues were homogenized, and the total RNA was extracted using a RNeasy mini kit (Qiagen).

## Quantitation of SARS-CoV-2 RNA copy numbers

SARS-CoV-2 RNA was detected using the nCoV_IP2 primer-probe set (National Reference Center for Respiratory Viruses, Institute Pasteur) which targets the SARS-CoV-2 RdRP gene. RT-qPCR reactions were performed using an Applied Biosystems 7500 real-time PCR system using the StepOnePlus 2.1 real-time PCR package. The nCoV_IP2 primer-probe set (nCoV_IP2-12669Fw ATGAGCTTAGTCCTGTTG; nCoV_IP2-12759Rv CTCCCTTTGTTGTGTTGT; nCoV_IP2-12696bProbe(+) 5′FAM-AGATGTCTTGTGCTGCCGGTA-3′BHQ-1) was used in combination with TaqMan fast virus 1-step master mix (ThermoFisher Scientific) to detect viral RNA. A standard curve was created using a PCR fragment (nt 12669-14146 of the SARS-CoV-2 genome) generated from viral complementary DNA using the nCoV_IP2 forward primer and the nCoV_IP4 reverse primer to quantitate the RNA copy numbers. RNA copy numbers were normalized to the weight of tissues used.

## SARS-CoV-2 genome sequencing

SARS-CoV-2 positive specimens were sequenced using metagenomic next-generation sequencing[40], Swift Biosciences SNAP panel[41], and/or Illumina COVIDSeq following the manufacturer's protocol. Sequencing reads were analyzed and visualized using cutadapt 1.9, bwa version 0.7.17, Picard 2.18.15, VarScan 2.3, Annovar 2018Apr16 version for the Longitudinal Analysis of Viral Alleles (LAVA; available at https://github.com/michellejlin/lava) pipeline with references MZ433225.1 for VOC gamma, NC_045512.2 for VOC delta, and OL965129.1 for VOC omicron[42]. Sequencing reads are available in NCBI BioProject PRJNA803552 (https://www.ncbi.nlm.nih.gov/bioproject/?term=PRJNA803552).

## Statistics and reproducibility

The Microsoft Excel (versions 16.52) and Numbers (version 10.1) software packages were used for most data collection. The GraphPad Prism (version 9.1.0) software package was used for data analysis. Reverse transcription RT-qPCR data were collected and analyzed using the StepOnePlus (version 2.1; Applied Biosystems) software package. Figures were assembled using Adobe Illustrator (version CS6). Power analyses were carried out using GPower 3.1. T-tests were used to evaluate statistical significance between experiments with two sets of data. One- and two-way ANOVAs with Dunnett's, Tukey's, or Sidak's comparisons post-hoc tests without further modifications were used to evaluate statistical significance when more than two groups were compared or datasets contained two independent variables, respectively. Specific statistical tests are specified in the figure legends for individual studies. The Supplementary Dataset 1 summarize all statistical analyses (effect sized, $P$ values, and degrees of freedom), respectively. Effect sizes between groups were calculated as $\eta^2 = \frac{SS_{effect}}{SS_{total}}$

for one-way ANOVA and $\omega^2 = \frac{SS_{effect} - (df_{effect}) \times (MS_{error})}{MS_{error} + SS_{total}}$ for two-way ANOVA; $df_{effect}$, degrees of freedom for the effect; $SS_{total}$, sum of squares for total; $SS_{effect}$, sum of squares for the effect; $MS_{error}$, mean square error. Effective concentrations for antiviral potency were calculated from dose-response datasets through four-parameter variable slope regression modelling. A biological repeat refers to measurements taken from distinct samples, and the results obtained for each individual biological repeat are shown in the figures along with the exact size ($n$, number) of biologically independent samples, animals, or independent experiments. The measure of the center (connecting lines and columns) is the mean throughout unless otherwise specified. The statistical significance level ($\alpha$) was set to <0.05 for all experiments. Exact P values are shown in Supplementary Dataset 1 and in individual graphs when possible.

### Reporting summary

Further information on research design is available in the Nature Research Reporting Summary linked to this article.

## Data availability

The metagenomic sequencing reads generated in this study have been deposited in the NCBI BioProject database under accession code PRJNA803552. All other data generated in this study are provided in this published article and the Supplementary Information/Source Data file. Source data are provided in this paper.

## Code availability

Metagenomic sequencing reads were analyzed and visualized using the LAVA pipeline[42] (available at https://github.com/michellejlin/lava). All commercial computer codes and algorithms used are specified in the Methods section.

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

## Acknowledgements

We thank members of the GSU High Containment Core and the Department for Animal Resources for support, and A.L. Hammond for critical reading of the manuscript. This work was supported, in part, by Public Health Service grants AIO71002 (to R.K.P.) and AI141222 (to R.K.P.), from the NIH/NIAID. The funders had no role in study design, data collection, interpretation, or the decision to submit the work for publication.

## Author contributions

C.M.L., R.M.C., and R.K.P. conceived and designed the experiments. C.M.L., R.M.C., J.S., J.D.W., and R.K.P. conducted most of the experiments. K.J, Q.P., and A.L.G. performed next-generation sequencing. M.T.S. performed chemical synthesis. M.S., Z.S., and A.A.K. performed mass spectrometry analysis. M.G.N. and G.R.P. provided critical materials. K.S. performed histopathology of organ samples. C.M.L., R.M.C., J.S., A.L.G., and R.K.P. analyzed the data. R.M.C. and R.K.P. wrote the manuscript.

## Competing interests

M.G.N. and G.R.P. are coinventors on patent 20190022116, $N^4$-Hydroxycytidine and derivatives and anti-viral uses related thereto, covering composition of matter and method of use of EIDD-2801 for antiviral therapy. This study could affect their personal financial status. All other authors declare no competing interests.
