## [Peer Review File · Nature Communications]

REVIEWER COMMENTS

Reviewer #1 (Remarks to the Author):

I want to congratulate the authors on this nice paper, which clearly reflects A LOT of work. The assays are well executed and the manuscript is well written. The authors use the different models for SARS-CoV-2 that are currently available in the field, both ex vivo as well as in vivo. They find that Molnupiravir (parent NHC) has a strong antiviral effect both in cell culture as well as in HAE system, which holds true for all the Variants of Concern. Furthermore, in ferrets Molupiravir reduces shedding titers in nasal lavages for ferrets infected with all VOCs, and subsequently suppresses transmission to untreated sentinel ferrets. Secondly, they used dwarf hamsters as a lethal COVID-19 model to determine infectivity, pathogenicity lethality and viral tissue tropism of each VOC, which is an interesting study. Also, in this model, Monupiravir is able to rescue infected hamster, independent of VOC type. Interestingly, while lung viral load is decreased in all groups, the magnitude of the effect varies depending on the infected VOC. They claim this difference can be due to sex differences. Furthermore, they inspect viral samples of the hamsters for emergence of resistance, in which they did not find any specific molnupiravir treatment-caused mutations. Lastly, after further inspection of the lung tissue of infected animals, show VOC dependent issues but Molnupiravir alleviated histopathology associated with either VOC.

This is a very extended paper in comparing the effect of infection with the different VOCs and subsequently the effect of Molnupiravir treatment. However, I think that the novelty mainly lies in the sex difference that was observed with treated dwarf hamsters that were infected with Omicron. Unfortunately, I feel that this part misses some extra studies of confirmation before such a strong claim can be made. Further, I find some unexplained inconsistencies that were left unexplained.

These are my questions for revision:

- Please explain why you think NHC is more effective against the variants compared to the original strain (EC50 difference). Was the same MOI used for the VeroE6 experiment?

Are the differences in viral titer that are observed in the vehicle treated groups related to the differences of infectivity in animal models? (eg. Clearly higher titer in gamma vs delta).

- please formulate EC50s with a SD or as in IQ range.

- Are the HAE-ALI experiment done by pre-treatment?

- If you stop treatment in the ALI studies, do you know if viral replication starts again? (from any possible residual virus, even though indetectable)? Only then you can speak from a sterilizing effect.

It is mentioned that VOC Beta did not give a productive infection in ferret, is it known why?

- Besides Delta, also Gamma does not show any infectious particles in nasal lavage after treatment with Molnupiravir at any time, so I don't understand why only Delta is excluded from transmission studies.

- How can you explain that effect size of Molnupiravir in viral load reduction is only different between VOCs in dwarf hamster, and not in vitro or in ALI experiments and not in ferret?

- How can you explain that biological sex only makes a difference when animals are infected with Omicron variant. Since this is the main novelty of the paper, I think it should be confirmed in other animal models.

Reviewer #2 (Remarks to the Author):

This study by Lieber et al reports the use of dwarf hamsters as a model for severe COVID-19 and assessing the efficacy of molnupiravir. The authors utilize several SARS-CoV-2 variants of concern (VOC) and report some intriguing findings on variability between the different variants.

They initially examine the antiviral potency of molnupiravir against the alpha, beta, gamma, delta, and omicron variants in VeroE6 cells and differentiated human airway epithelial cells. No substantial differences were observed, with molnupiravir being equally effective against all variants. They next used the ferret model (to assess transmission), and although no differences were seen with molnupiravir efficacy, they report that the Beta variant didn't infect ferrets, and that Omicron didn't transmit (probably linked to the much lower titers in nasal lavage). This is a good example of where we can question the accuracy of animal models, as we know that in humans Omicron is the most transmissible variant to date.

The authors then moved on to the Roborovski dwarf hamster model, which unlike the Syrian hamster, provides a lethal model for SARS-CoV-2 infection. These data are considered to be the most novel of those presented in this study. Comparison of the VOCs indicate that Delta is the more pathogenic in this model, and Alpha is the least pathogenic (80% of animals survived). The authors claim that this reflects the severity of disease in humans (citing the difference between delta and omicron), but I think more caution is required. Disease severity in humans is obviously impacted by immune status, and because the variants arose at different times, the immune status of the human population was not comparable (as it is with an animal). Regardless, molnupiravir was able to protect against lethal infection by any VOC. However, there was variation in the viral load reduction. The authors propose that this could be due to the differential spread of the virus to organs other than the lung, and potential impacts on the pharmacokinetic properties of molnupiravir. This seems plausible, and will have to be investigated further. Perhaps the most surprising (or startling) data are those showing that molnupiravir is significantly less effective in female dwarf hamsters – but only against omicron. Much like the authors, I have no explanation for this!

As with all animal models, we are left to consider whether this dwarf hamster model accurately reflects infections in humans. In particular, the significance of the sex bias in molnupiravir efficacy (only with omicron), is something that we should be alert to in clinical data emerging during the omicron wave. Overall, I think this is an intriguing study that perhaps doesn't so much provide answers, as questions. Nevertheless, the experiments are thorough, and the questions raised will help to advance the field.

Response to reviews:

Reviewer #1

I want to congratulate the authors on this nice paper, which clearly reflects A LOT of work. The assays are well executed and the manuscript is well written. [...] This is a very extended paper in comparing the effect of infection with the different VOCs and subsequently the effect of Molnupiravir treatment. However, I think that the novelty mainly lies in the sex difference that was observed with treated dwarf hamsters that were infected with Omicron. Unfortunately, I feel that this part misses some extra studies of confirmation before such a strong claim can be made. Further, I find some unexplained inconsistencies that were left unexplained. These are my questions for revision:

We thank the reviewer for their appreciation of data quality, presentation of the work, and importance of our study. We concur that the observation of sex differences in the response to molnupiravir treatment of VOC omicron infection is entirely novel. However, this finding is far from the only novelty of our work as implied in the review. Rather, in this study we establish Roborovski dwarf hamsters as a novel drug efficacy model to test the effect of SARS-CoV-2 inhibitor on acute, life-threatening viral pneumonia. This acute lung injury model closes a major gap impairing COVID-19 drug development, since none of the current animal models (mice, transgenic mice, Syrian golden hamsters, ferrets) recapitulates COVID-19 respiratory failure with acute lung injury. Pioneering a much-needed drug efficacy model thus constitutes a major advance of the field.

1) Please explain why you think NHC is more effective against the variants compared to the original strain (EC50 difference). Was the same MOI used for the VeroE6 experiment?

Earlier dose-response experiments testing NHC against the SARS-CoV-2 WA-1 strain were carried out in Vero-E6 cells, whereas we used the more recently generated Vero-E6 cells stably expressing TMPRSS2 (Vero-TMPRSS2) in this study. To facilitate direct comparison of the original SARS-CoV-2 isolate and the VOC, we have retested NHC against WA-1 and show the resulting dose-response curve in a revised Fig. 1A. This study revealed that under equal experimental conditions NHC is equally effective against WA-1 and the VOC tested. We have revised the description of this experiment in the Results section accordingly.

2) Are the differences in viral titer that are observed in the vehicle treated groups related to the differences of infectivity in animal models? (eg. Clearly higher titer in gamma vs delta).

Ex vivo, we noted pronounced differences in VOC gamma vs delta titers in the HAE cultures (Fig. 1B), but not in Vero-TMPRSS2 cells (Fig. 1A). Differences in shed VOC gamma vs delta peak titers in ferrets were not statistically significant (revised Fig. 2B, now showing results of a 2-way ANOVA with Tukey's post-hoc test comparing VOC gamma and delta) and VOC gamma vs delta lung titers in Roborovski dwarf hamsters were virtually identical (Fig. 2C) in vehicle treated animals. The experimental results thus indicate that differential VOC gamma vs delta replication in the HAE cultures does not translate into differences of infectivity in the animal models.

3) please formulate EC₅₀s with a SD or as in IQ range.

We have revised Fig. 1A, the only figure in which EC₅₀ values were calculated. The revised figure now shows EC₅₀ values and 95% confidence intervals. Calculating SD is not compatible with the regression modeling approach applied to these datasets.

4) Are the HAE-ALI experiment done by pre-treatment?

We have clarified that the inhibitor was added to the basolateral chamber of the transwell at the time of apical infection of the cultures.

5) If you stop treatment in the ALI studies, do you know if viral replication starts again? (from any possible residual virus, even though indetectable)? Only then you can speak from a sterilizing effect.

This point is very well taken. In response, we have revised the wording describing this result to avoid misunderstandings. We intended to convey that 10 μ M basolateral NHC blocked all virus replication in the HAE system, and the revised wording better communicates this observation. It is indeed conceivable that virus replication may restart in HAE cultures if compound was removed. We have not experimentally explored this possibility, however, since it lacks physiological relevance. In contrast to the *in vivo* situation, the HAE cultures do not contain lymphocytes. Whereas pharmacological block of virus replication may indeed be reversible in the absence of innate immune cells, infiltrating lymphocytes would trigger local inflammation and viral clearance *in vivo*. Please also note that in contrast to recent reports describing rare recurrence of COVID clinical signs after a paxlovid regimen, no such virus rebound has been described for patients on molnupiravir.

6) *It is mentioned that VOC Beta did not give a productive infection in ferret, is it known why?*

This result was surprising to us also and the reason is not immediately clear, especially since RBD mutations in spike of VOC beta (no productive infection of ferrets) and gamma (productive infection of ferrets) are identical. We authenticated the virus inoculum before infection of ferrets and found genome integrity to be intact. Passage of SARS-CoV-2 through ferrets results typically in rapid emergence of weasel-characteristic mutations (mentioned also in the manuscript), but there is no obvious explanation for why such species-adaptation did not occur in the case of VOC beta. It appears that there was very little/no VOC beta replication in ferrets at all, which cannot have been due to quality of the virus inoculate, since the same stock was used successfully for the subsequent dwarf hamster study.

7) *Besides Delta, also Gamma does not show any infectious particles in nasal lavage after treatment with Molnupiravir at any time, so I don't understand why only Delta is excluded from transmission studies.*

This is a case of hindsight is 20/20. The VOC gamma study was carried out first and we decided at the time to test molnupiravir efficacy and effect on suppressing virus transmission in parallel rather than in consecutive studies. Based on complete suppression of VOC gamma replication by molnupiravir and lack of transmission, however, we designed the subsequently conducted evaluation of molnupiravir activity against VOC delta as an efficacy study only to minimize the number of large research animals required for the project as much as possible. Since treatment completely suppressed VOC delta shedding in nasal lavages (mimicking the results of the earlier VOC gamma study), we considered it unethical to invest additional ferrets for a VOC delta transmission study.

8) *How can you explain that effect size of Molnupiravir in viral load reduction is only different between VOCs in dwarf hamster, and not in vitro or in ALI experiments and not in ferret?*

The strongest infection model is the one that most closely reproduces the clinical phenotype. In the case of molnupiravir, we are in the strong position that Merck has publicly released a plethora of clinical data. As referenced in the manuscript, patients infected with VOC gamma experienced a greater benefit from molnupiravir treatment in clinical trials than those suffering from VOC delta. The Roborovski dwarf hamster model reproduced this VOC-specific difference in treatment benefit, other models tested did not.

We fully agree with the reviewer that it is puzzling why cell culture, HAE organoid, and ferret models failed to recapitulate the difference in benefit seen in human patients. Candidly, we do not know the reason. Certainly, none of the available *ex vivo* and *in vivo* SARS-CoV-2 infection models fully reproduce human disease, and each has distinct strengths and weaknesses. Careful model selection continues to be required to ensure relevance of experimental data. That is why we consider broadening the panel of available experimental systems through the

development of the Roborovski dwarf hamster drug efficacy model an important advance of the field.

9) *How can you explain that biological sex only makes a difference when animals are infected with Omicron variant. Since this is the main novelty of the paper, I think it should be confirmed in other animal models.*

We feel that pioneering a drug efficacy model that reproduces respiratory failure of severe COVID-19 and recapitulates, in the case of VOC gamma and delta, clinical trial results is a major novelty. It is less clear to us, however, what confirmation other animal models might provide. The experience with VOC gamma and delta (outlined above in the response to point #8 of this reviewer) highlights that outcome in a majority of models does not necessarily ensure greater predictive power for human disease.

Aggravating the conundrum, no candidate confirmatory other animal model appears to exist. It could not be ferrets, as we have shown in this study. In mice, VOC beta and gamma replicate in our experience to high lung virus load, albeit without any clinical signs. However, VOC delta fails to replicate in mice, and VOC omicron reportedly replicates poorly (PMID 35062015). VOC omicron reportedly (PMID 35062015) replicates poorly also in Syrian golden hamsters, reaching lung virus loads that are approximately 3-4 orders of magnitude lower than what we observed in the Roborovski dwarf hamsters. Due to the resulting narrow dynamic range of the Syrian golden hamster model for VOC omicron, multi-tier differences in treatment effect size could not be appreciated in Syrian goldens. This leaves non-human primates, which we would not commit to this study for ethical reasons.

The Roborovski dwarf hamster results shown in this manuscript are supported by considerable statistical power. We have taken great care to caution that at present we do not know whether the VOC omicron data (sex difference of molnupiravir treatment benefit) equally translate to the human situation. Rather, the dwarf hamster model alerts us of the possibility, raising awareness that benefit of available COVID-19 medicines must be reassessed continuously as new VOC emerge. We stand by this conclusion, which does not require support by other models.

Reviewer #2

This study by Lieber et al reports the use of dwarf hamsters as a model for severe COVID-19 and assessing the efficacy of molnupiravir. The authors utilize several SARS-CoV-2 variants of concern (VOC) and report some intriguing findings on variability between the different variants.

[...]

1) The authors then moved on to the Roborovski dwarf hamster model, which unlike the Syrian hamster, provides a lethal model for SARS-CoV-2 infection. These data are considered to be the most novel of those presented in this study. Comparison of the VOCs indicate that Delta is the more pathogenic in this model, and Alpha is the least pathogenic (80% of animals survived). The authors claim that this reflects the severity of disease in humans (citing the difference between delta and omicron), but I think more caution is required. Disease severity in humans is obviously impacted by immune status, and because the variants arose at different times, the immune status of the human population was not comparable (as it is with an animal).

This is a fair point. We have revised the statement in the first paragraph of the Discussion section, now solely focusing on differential pathogenicity of individual VOC in the Roborovski dwarf hamster model.

2) Regardless, molnupiravir was able to protect against lethal infection by any VOC. However, there was variation in the viral load reduction. The authors propose that this could be due to the differential spread of the virus to organs other than the lung, and potential impacts on the pharmacokinetic properties of molnupiravir. This seems plausible, and will have to be investigated further.

We agree, and have conducted a pharmacokinetic (PK) study for this revision that evaluates NHC plasma concentrations and, at 3 time points after dosing, tissue distribution of the bioactive triphosphate form NHC-TP. No significant differences in PK profiles and tissue exposure were

detected between male and female Roborovski dwarf hamsters. We have included these additional data in a new Figure 5 and new Supplementary Table 2 in the revised manuscript. Former Figure 4 has been split into revised Figure 4 and Figure 6, respectively, to preserve flow of the presentation of results. All figure pointers in the text were updated accordingly.

3) As with all animal models, we are left to consider whether this dwarf hamster model accurately reflects infections in humans. In particular, the significance of the sex bias in molnupiravir efficacy (only with omicron), is something that we should be alert to in clinical data emerging during the omicron wave. Overall, I think this is an intriguing study that perhaps doesn't so much provide answers, as questions. Nevertheless, the experiments are thorough, and the questions raised will help to advance the field.

Thank you very much for this most insightful overall evaluation of our work. We indeed believe that the greatest impact of this study is the reveal of knowledge gaps in the treatment dynamics of severe COVID-19 caused by different VOC. The work shows that predicting unaltered drug efficacy against different VOC based on unchanged *ex vivo* inhibitory concentrations misses the complexity of the dynamic host-pathogen-antiviral drug interplay. Appreciating these differences in the Roborovski dwarf hamster model alerts to the need for continued vigilance in the clinic to ensure that patients will receive medicines that provide them with the best risk/benefit ratio.

REVIEWERS' COMMENTS

Reviewer #1 (Remarks to the Author):

The authors addressed all my comments and questions with care, thank you for that.

Although I'm not convinced of the sex-effect (or rather, I am still longing for some more explanation, validation, ...) the data that is presented is solid and clear. The manuscript can be accepted.

Reviewer #2 (Remarks to the Author):

The authors have thoroughly addressed all points raised by the reviewers and the addition of the revised data further strengthens this study. I have no other concerns.

Response to reviews:

Reviewer #1

The authors addressed all my comments and questions with care, thank you for that. Although I'm not convinced of the sex-effect (or rather, I am still longing for some more explanation, validation, ...) the data that is presented is solid and clear. The manuscript can be accepted.

Thank you very much for this favorable overall assessment of our study.

We are aware that it is currently not possible to predict whether the biological sex-driven difference in antiviral effect size seen in the dwarf hamsters will equally apply to the human host. We had included a statement to this regard in the Discussion section. Independent from this question, our work highlights that the dynamic host-SARS-CoV-2-antiviral drug interplay is complex and VOC-specific. In this final revision, we have expanded our statement, now better emphasizing that the Roborovski dwarf hamster model illuminates that VOC-specific differences in treatment effect size may be present *in vivo*. This observation calls for continued vigilance in the clinic as new VOC emerge to ensure that patients will receive medicines that provide them with the therapeutic best benefit.

Reviewer #2

The authors have thoroughly addressed all points raised by the reviewers and the addition of the revised data further strengthens this study. I have no other concerns.

Thank you very much for this unconditional endorsement of our work. We are grateful for your insightful comments that have helped us to strengthen the study.